# Developing Recommendations for Cumulative Endpoints and Lifetime Use for Research Animals

**DOI:** 10.3390/ani11072031

**Published:** 2021-07-07

**Authors:** Elizabeth A. Nunamaker, Shawn Davis, Carly I. O’Malley, Patricia V. Turner

**Affiliations:** 1Animal Care Services, University of Florida, 1600 Archer Rd, Gainesville, FL 32610, USA; nunamaker@ufl.edu; 2Animal Care Services, Brock University, 1812 Sir Isaac Brock Way, St Catherines, ON L2S 3A1, Canada; nb_spetrik@brocku.ca; 3Global Animal Welfare and Training, Charles River Laboratories, Wilmington, MA 01887, USA; 4Department of Pathobiology, University of Guelph, Guelph, ON N1G 2W1, Canada

**Keywords:** cumulative use, humane experimental endpoints, alternatives, animal welfare, laboratory animals, 3Rs

## Abstract

**Simple Summary:**

The welfare of research animals should be a top priority in any research program, and that includes their long-term welfare. Animals may be used in multiple experiments or used for training purposes, which may lead to cumulative suffering. To prevent this, humane endpoints need to be defined specifying limits on experiments, procedures, and time that animals are used for research. There are few resources available for deciding which criteria to use when making endpoint decisions. The purpose of this paper is to present results of a survey identifying laboratory animal professionals’ attitudes and institutional strategies regarding cumulative endpoints, review regulations and tools addressing cumulative welfare in research animals and provide recommendations for how to move forward addressing this issue. While institutions may have endpoint guidelines in place, many of them only cover certain species or may be informal. There is a need for more specific guidelines that allow for the diversity of experiments using animals. Welfare assessment tools could help provide objective guidance on humane endpoints for research animals. Further research on which tools are the most efficient and comprehensive would be beneficial in improving animal welfare as well as the quality of the science using research animals.

**Abstract:**

Research animals are important for scientific advancement, and therefore, their long-term welfare needs to be monitored to not only minimize suffering, but to provide positive affective states and experiences. Currently, there is limited guidance in countries around the world on cumulative and experimental endpoints. This paper aims to explore current opinions and institutional strategies regarding cumulative use and endpoints through a scoping survey and review of current regulations and welfare assessment tools, and ultimately to provide recommendations for assessment of cumulative and lifetime use of research animals. The survey found that only 36% of respondents indicated that their institution had cumulative use endpoint policies in place, but these policies may be informal and/or vary by species. Most respondents supported more specific guidelines but expressed concerns about formal policies that may limit their ability to make case-by-case decisions. The wide diversity in how research animals are used makes it difficult for specific policies to be implemented. Endpoint decisions should be made in an objective manner using standardized welfare assessment tools. Future research should focus on robust, efficient welfare assessment tools that can be used to support planning and recommendations for cumulative endpoints and lifetime use of research and teaching animals.

## 1. Introduction

Research animals are vital for the advancement of science; however, consideration of their long-term welfare is a continuing concern. The goal of implementing the 3Rs (replacement, reduction and refinement) as an animal welfare framework [1] is to minimize the suffering of research animals, and increasingly, to promote positive affective states and experiences [2]. To ensure that animals do not suffer when being used as biomedical models, humane endpoints should be defined in the study protocols [3,4,5]. Humane endpoints have been defined as “the earliest indicator in an animal experiment of pain and/or distress that, within the context of moral justification and scientific endpoints to be met, can be used to avoid or limit pain and/or distress by taking actions such as humane killing or terminating or alleviating the pain and distress” [6]. 

Not all scientific endpoints in animal-based research require euthanasia and thus many animal ethics committees (AEC) have adopted an alternative term to describe animal disposition when the scientific endpoint is reached, the experimental endpoint. An experimental endpoint in this context may include euthanasia or provision of analgesia, but could also include repurposing the animal, removal from the study, providing a test article ‘holiday’, implementing designated rest periods between studies, retirement of the animal from research or adoption [3]. Reuse of animals in multiple studies or procedures may result in cumulative suffering due to repeated performance procedures that may cause pain, discomfort or stress [7]. Along a similar line of reasoning, cumulative endpoints and overall lifetime use should be considered for animals used in one or more protocols for an extended period (i.e., duration of use) or in individual protocols that involve multiple procedures conducted over an extended period of time (i.e., multiplicity of procedures, frequency of use, and intensity of use) when euthanasia is not required at study end [8]. Many protocols also may not take into account errors or variation in skill level of those conducting techniques on animals that may result additional procedures (for example, multiple venipuncture attempts to obtain a single sample). Any of these potential harms may be intensified for animals maintained in less than optimal environments. Literature concerning cumulative endpoints is scarce and there is no regulatory information available, therefore guidance on what criteria to use to make endpoint decisions in these situations is difficult and often left to the discretion of the researcher or institutional AEC. The use of animals in multiple studies is an issue that many AECs struggle with, particularly for sensitive or difficult to acquire species.

This paper aims to explore current opinions, strategies to assess cumulative welfare impact and experimental animal endpoints by Animal Ethics Committees, and what institutional solutions might be in place for which species. Summary results will be presented from a scoping survey that was administered to laboratory animal professionals to determine current attitudes towards and institutional approaches for cumulative use guidelines and management of research animals. Current regulations pertaining to humane endpoints for research or teaching animal will be discussed, followed by an overview of tools available for assessing the welfare of research animals. Finally, recommendations will be provided for institutional assessment of cumulative and maximum lifetime use for research animals to encourage greater consideration of the ethical implications of non-terminal research or training and educational use of animals. These recommendations are applicable wherever animals are worked with for research, teaching, and testing.

## 2. Scoping Survey about Frequency of Use, Lifetime Use, and Cumulative Use Considerations for Research Animals

### 2.1. Survey Methods 

A questionnaire was created to gather information from global laboratory animal professionals regarding institutional policies and procedures for considering lifetime use and cumulative endpoints for research and teaching animals. No identifying information was collected of participants and the survey results were accumulated by Canadian Association for Laboratory Animal Medicine (calam-acmal.org) association management personnel, with only aggregate data provided to the researchers. This study was deemed exempt for REB approval because of this. Participants were informed before answering the survey that their participation was voluntary, all answers would be anonymous, no incentives were used, and that they could choose to leave questions unanswered at their discretion. 

The questionnaire was developed by the research team and entered into SurveyMonkey (SurveyMonkey Inc., San Mateo, CA, USA). The survey consisted of 17 questions divided into 6 sections including Part A: demographics of respondents; Part B: current policies or procedures for experimental endpoint decision making; Part C: lifetime use of animals in research and teaching; Part D: quality of life and lifetime use endpoints; Part E: reduction vs. refinement; and Part F: concluding questions asking participants for their final thoughts and comments regarding lifetime use and cumulative endpoints for research and teaching animals (see Appendix A). The survey questions are provided in Figure 1. The questionnaire was open 14–28 February 2017.

To participate in this study, individuals had to be 18 years of age or older, have worked with research and teaching animals, and be members of the CompMed (*n* = ~10,000 members) or European College of Animal Welfare and Behavioural Medicine (ECAWBM) (*n* = ~150 members) list serves. The research team members did not participate in the survey.

### 2.2. Data Analysis

Data are presented as percentage of responses out of the total number of respondents for each question. Because only aggregate responses were collected for each question, it was not possible to perform more detailed statistical analyses. Open-ended responses were later manually entered into Excel ver 2019 16.0 (Microsoft Corp, Redmond, CA, USA), where common answers were summarized. Some quotes were edited for clarity and conciseness, but overall maintained the original message provided in the survey.

### 2.3. Results

#### 2.3.1. Respondent Demographics

The survey yielded 154 responses, ranging across geographic regions. The results of the demographic questions of the survey are presented in Table 1. Most respondents were from North America (59.7%), then the European Union (24.0%). The age range of respondents was well distributed from 26 to 65 years of age, and most respondents were female (66.7%). Over half of the respondents worked at an academic institution (60.5%). Over half (58.2%) of the respondents were veterinarians and the primary species worked with were mice (77.7%) and rats (68.8%).

#### 2.3.2. Institutional Endpoint Policy and Disposition of Animals at Study Conclusion

Most respondents (77%) indicated that their AEC had a formal endpoint policy in place, while 23% indicated that they did not. Mice (87%) and rats (86%) were most likely to be covered by a formal endpoint policy (likely because these were the most common animals reported in the respondents’ facilities), with rabbits (72%) also typically covered. For all other species, at least 30% of respondents indicated that those animals were covered as well.

Events most likely to trigger an endpoint discussion were the presence of an acute condition resulting in moribund state (95%), deteriorating quality of life due to existing health or behavioral conditions (94%), level of invasiveness of the research protocols (70%), number of procedures conducted on the animal (i.e., cumulative lifetime use; 68%), and deteriorating quality of life because of existing social conditions (54%). Endpoint decisions other than euthanasia are summarized in Figure 1. The most common alternative to euthanasia was adoption (59.8%), which was most often considered for dogs, rabbits, and cats (Figure 2a). Livestock species were more likely to be sent to slaughter at study endpoint (Figure 2b), returned to the source they were acquired from (5.7%), sold (3.4%) or given to a farm (3.4%). Primates were sometimes sent to a sanctuary (6.9%) and wildlife, including bats and sparrows, were released back into the wild (6.9%).

#### 2.3.3. Quality of Life and Lifetime Use Decision Making

Participants were asked their opinion on specific scenarios related to lifetime use of research and teaching animals. The scenarios and the responses are summarized in Figure 3.

How institutions make decisions regarding quality of life and lifetime use endpoints is presented in Table 2. Some respondents indicated that quality of life and endpoint decisions were outlined in individual study operating procedures or the animal use protocol, and thus the decisions may not be made by the institution (in the EU, the ethical review is not done at the institutional level). Others indicated that endpoints are decided on a case-by-case basis, based on clinical signs or that their institution did not have long-term housed animals. One respondent outlined their quality of life and lifetime use endpoint protocol as:

“Animals must not have more than 2 major survival surgical procedures, unless heavily justified scientifically. If more blood needs to be drawn than specified in the guidelines, it must be a terminal procedure. An animal may not be on more than one “Category E” protocol (i.e., potential for severe suffering) unless it was a control animal. Aged animals on training protocols should be used first for terminal procedures unless there is a scientific justification. Any animal exhibiting stress from study procedures must be removed from that study or the study protocol that requires that procedure by assignment to another protocol, returned to vendor, used as control, used as companion animal, blood donor, or similar, or euthanized. All moribund animals will be euthanized (use score sheet and/or by veterinarian determination)”.

#### 2.3.4. Consideration of Cumulative Use of Animals—Refinement vs. Reduction

To gain a better understanding of opinions on the importance of reduction versus refinement, survey respondents were given a hypothetical scenario. Specifically, in a long-term study requiring over 100 blood collections per animal without the use of a catheter or vascular access port, survey respondents preferred refinement over reduction. More respondents felt that using more animals with fewer repeated samples (56%) was preferable to fewer animals and multiple sample collections (9%). In response to “Other” (35%), respondents indicated that their answer would depend on the study, species, age of the animal, study duration, whether the animal was sedated or habituated to blood collection or other criteria. Some said they would only use a catheter or vascular access port for this type of study. Respondents also stated that refinement to the experimental design or study protocol was needed.

In another scenario describing a long-term study that required >10 liver biopsies obtained via percutaneous ultrasound-guided biopsy under general anesthesia, refinement was again preferred over reduction. Most respondents indicated that they would use more animals with fewer repeat sample collections (52%). Some (23%) said they would use fewer animals with multiple repeat sample collections. In response to “Other” (25%), it was again emphasized that the answer was dependent on several criteria and that refinement of the study protocol was needed.

When asked what consideration was most appropriate for a colony of animals used in repeated, minimally invasive studies that required multiple sample collections, refinement was preferred over reduction. Most (69%) respondents felt there should be a limit to the total number of studies that an animal was used in. In contrast, a small number favored reduction, indicating no limit on study number was needed, but the total number of animals used should be minimized (9%) or indicated no limit is needed because there was no welfare concern (6%). Those that selected “other” (16%) indicated that their answer would depend on several criteria including the type and frequency of collections being done, or indicated that stress should be minimized, positive reinforcement training should be used, and/or that animal welfare needed to be regularly evaluated.

Survey respondents were asked whether their answers to the previous questions would change if the animals needed to be single housed for 1–2 weeks during the study. The majority said no (69%). Those that said yes suggested that fewer animals should be used (11%), or alternatively, more animals should be used to decrease the distress for an individual animal (6%). ‘Other’ responses (14%) indicated that it would depend on the species, housing structures, and the cumulative number of single-housing periods, that single housing was not a concern for that short a timeframe, and that additional enrichment or visual contact with conspecifics was needed to minimize negative impacts of single housing.

#### 2.3.5. Interest in Cumulative and Lifetime Use Guidelines at an Institutional Level

Finally, most survey respondents (85%) indicated that they would support the adoption of species-specific guidelines at their institution. Respondents specified that broad policies regarding lifetime use and cumulative endpoints may be inappropriate and generally supported general guidelines that could be considered on a case-by-case basis. Some of the comments are provided below:“I think balance and good benefit/cost analysis is needed, we cannot create guidelines for a broad range of experiments and animal species. Each study needs a singular evaluation, discussion, and reflection from the Animal Welfare Body”.“I think blanket policies may be inappropriate, and instead there might need to be some general guidelines that can be applied on a case by case basis directly within protocols where all the details can be considered independently”.“I am always concerned that policies, while good as guidelines, may limit the ability for the veterinarian or IACUC to utilize professional judgment in deciding the fate of individual animals”.“The approach should be a whole animal assessment by a cross-disciplinary team”.“Principal investigators are strongly discouraged from advocating animal reuse as a reduction strategy, and reduction should not be a rationale for reusing animals or animals that have already undergone experimental procedures especially if the well-being of the animals would be compromised. My opinion is that you should strive for minimal suffering per individual animal even in the cost of using more animals”.“Committees should come up with objective criteria for the maximum number of lifetime procedures—similar to consideration for multiple major survival surgeries currently required by USDA—for a variety of procedures and apply those criteria to animals on a single study and then carefully weigh what should be allowable”.

### 2.4. Discussion

As issues of animal welfare are addressed for laboratory animals, there can be conflict between the 3Rs tenets, such that a reduction in animal numbers may lead to reuse of animals in multiple studies or procedures, resulting in cumulative suffering. Defining criteria for lifetime use and cumulative endpoints for research and teaching animals is an area that AECs the world over are currently struggling with. The aim of this survey was to determine whether and how lifetime use and cumulative endpoints are being tracked and evaluated at different institutions, and for which species.

While approximately two-thirds of respondents indicated that their institution had a formal experimental endpoint policy, only 36% said that the endpoint policies include criteria relating to lifetime use or cumulative experiences, such as number of studies conducted with the animal, number of study days, or total blood volume collected over the animal’s lifetime. Rats, mice, and rabbits were most likely to be covered under a formal endpoint policy. While rats and mice make up a significant portion of the research mammal population, endpoint decisions for other species can be complicated and need to be formally addressed.

Survey respondents largely supported the adoption of species-specific guidelines at their facility but expressed concern over formal policies. Respondents indicated multiple times throughout the survey that their thoughts about lifetime use and cumulative endpoints were dependent on details such as species, study procedures, animal age, housing, and training. Due to diversity of studies and institutional policies and procedures, it is likely only possible to generate overarching species-specific guidelines that could be implemented on a case-by-case basis. Alternatively, or in parallel, institutions may choose to provide limits for animals based on use, for example, imposing two-year limits for teaching or blood donor animals. As suggested by a respondent, to implement guidelines regarding cumulative endpoints, whole-animal assessments should be conducted by a cross disciplinary team. As part of the whole-animal assessments, a thorough harm-benefit analysis is needed for making endpoint decisions [9].

Respondents indicated that the use of positive reinforcement training to desensitize and habituate animals to study procedures would be an example of criteria that would influence their opinions on experimental endpoints. While desensitization and habituation of study animals are key factors in reducing stress during study procedures, limits on well-trained research and teaching animals should still be defined. As part of the survey, participants were asked their thoughts on the repeated use of a teaching mare for rectal examinations A little more than half of the respondents (54.5%) disagreed or strongly disagreed that the teaching mare be used twice weekly for her institutional life, while the remaining respondents were neutral or agreed that the level of lifetime use was acceptable. Criteria are needed to evaluate animal welfare throughout an animal’s institutional life based on their cumulative experiences, and to define experimental endpoints based on physiological, behavioral, psychological, and ethical guidelines [6].

Committees may need to work harder on end disposition of animals other than euthanasia. Other studies have suggested that the nature of consideration for animal use during a study changes if adoption or retirement are predetermined goals. The results of this survey indicate that cumulative use of animals in research or teaching settings is being considered by institutional AECs for at least some animals but further guidance would be helpful in this area. In the following sections of this paper, we outline currently available guidance, and review animal welfare indicators that can be used to determine more meaningful endpoints, and make recommendations for research organizations to establish better cumulative and lifetime use endpoints.

## 3. Relevant Cumulative Endpoint Guidance for Research Animals from Regulatory and Compliance Authorities

Although there is currently no specific regulatory guidance available in the USA, Canada, the UK or the EU for determining cumulative endpoints for research animals, there are activities required by one or more national and/or regional policymaking authorities that can be used to support decision making in this regard. This includes ethical review of the proposed research (alternatively known as harm: benefit analysis [HBA] in the UK [10] and the EU [11]), specific limits defined for specific procedures such as surgery, prospective and retrospective assessments of experimental severity, and humane endpoint guidance. Each of these topics will be discussed in further detail below in the context of how they may be used to support cumulative endpoints for research animals.

### 3.1. Ethical Review of Protocols by an AEC and/or Authority

Internationally, there is a common expectation [12] that all use of animals in science will undergo a priori ethical review or HBA (reviewed by [13]). The overall purpose of the ethical review is to weigh the costs of the proposed procedures against the potential harms or suffering that may be experienced by animals in the experiment. Although detailed approaches have been developed to conduct a rigorous HBA, in practice [14,15], when faced with hundreds of project reviews and amendments that need to be conducted in a timely fashion, many AECs must necessarily limit the scope of the ethical review. Typically, a 3Rs ethical framework [1] is applied, in which the reviewers evaluate information provided by the researcher on standardized protocol forms for the potential to replace animal use, reduce animal numbers, and refine procedures to minimize animal pain and distress [16]. This approach presupposes that animals will be used (and often the researchers have already been funded for the proposed body of work by national scientific granting agencies) and the work of the committee is to ensure that the work is conducted in an acceptable manner [17].

Areas of particular note for reviewers that pertain to cumulative endpoints include multiplicity of procedures, frequency of use, intensity of use, and duration of the protocol. Several hypothetical examples are provided in Table 3 that cover common areas of ethical concern for research and teaching animals.

### 3.2. Classifications of Major vs. Minor Surgical Procedures

Most regulatory authorities distinguish between major and minor surgical procedures (major procedures typically being those that penetrate a body cavity and/or ones that create a permanent physical or physiologic impairment). They then further discourage or restrict multiple major survival surgeries on a single animal within the same protocol or when an animal is transferred and used again in separate protocols. For example, the US Department of Agriculture (USDA) Animal Care Policy #14 specifically states that no animal may have more than one major survival operative procedure unless multiple procedures are required to meet the scientific objective of a single animal study activity, are scientifically justified by the PI, and approved by the AEC. This policy also states that major operative procedures that are part of the veterinary care program, and not research, do not count against this policy and there is no limit to the number of clinical care surgeries or procedures an animal can undergo [18]. This policy only applies to Animal Welfare Act-covered species, and excludes mice, rats, fish, etc.

Similarly, the Institute for Laboratory Animal Research (ILAR) Guide [4] recommends that multiple surgical procedures, major or minor, on a single animal, be evaluated to determine their impact on the animal’s well-being but provides no guidance on how to approach this. It further states that multiple major surgical procedures are only acceptable if they are: (1) included in and essential components of a single research project or protocol, (2) scientifically justified by the investigator, or (3) necessary for clinical reasons. The ILAR Guide also indicates that if multiple survival surgeries are approved, the AEC should pay particular attention to animal well-being through continuing evaluation of outcomes. The Canadian Council on Animal Care (CCAC) guidelines also strongly discourages the use of a single animal in multiple survival surgeries and indicates that multiple major surgery protocols must be approved by the institution’s AEC and allowed only if for scientific reasons. Multiple major surgeries on a single animal should not be done to save money, and a second major surgery may be performed if it is non-survival [19].

Despite these restrictions, it is commonplace in North America for scientists to secure AEC approval for surgical repairs, effectively creating a multiple survival surgery situation. This might be to repair telemetry or vascular access instrumentation in long-term experimental animals and the goal is to protect a scarce resource and minimize animal numbers in research, but this potentially comes at the cost of increased pain and stress for the individual animals. This is especially concerning given that pain is often under-recognized and treated following surgical procedures in research animals [20,21].

Some have called for a re-evaluation of the definition of major operative procedure. They propose that a major surgery using good peri-operative care and analgesia has a lesser negative impact than protocols that may involve multiple superficial insults, such as skin biopsies or repeated anesthesia events for imaging, which may be associated with a higher degree of welfare impact [22]. These discussions emphasize that labeling and/or limiting operative procedures does not always reduce cumulative harms for animals in specific protocols.

### 3.3. Prospective Assessment of Severity/Invasiveness of Procedures and/or Protocol

Consideration of the invasiveness of individual procedures and of the overall study plan is an important means of limiting cumulative severity of a project. Using this approach, the procedures to be conducted are classified according to the potential for pain and distress. In general, procedures that will be more painful generally require closer observation of the animal and may require development of a specific assessment score sheet [23,24]. The AEC may choose to set an ethical limit as to how much suffering is permissible in an experiment, especially if the animal is not provided appropriate mitigation [25,26]. There can be challenges with this assessment, however, because the harms may not always be known or apparent for a particular procedure or animal model [27], for example, a newly created line of genetically modified mice. Despite this, the prospective evaluation for severity does help to alert the research team and vivarium personnel to potential challenges that may arise within a study that may necessitate treatment of some kind or early removal of the animal from the study [11,28,29].

### 3.4. Retrospective Assessment of Actual Severity of Procedures and/or Protocol

Predetermining the potential for invasiveness in research is important for establishing scientific and humane endpoints, but it is also important to reflect on what the actual study outcomes or impacts of the procedures were. This is a requirement for at least some studies in the UK and the EU. Retrospective assessment can be very helpful for the AEC to better understand the potential risks for similar ongoing or future studies and it is typically conducted at the conclusion of the experiment or at the time of protocol renewal. In some cases, the prospective assessment may significantly underestimate the outcomes [9]. The retrospective assessment is an important exercise to look for further mitigations to reduce cumulative suffering of animals and can be part of general post-approval monitoring discussions with researchers. However, the retrospective assessment rarely covers operational issues that may impact cumulative suffering of animals, such as, multiple needle sticks required for a single sample, injuries due to conspecific fighting, higher rates of post-operative infection because of poor technique or environmental factors such as increased vibrations due to construction.

### 3.5. Establishment of Humane Endpoints for Research

The final area of protection afforded for research animals is establishment of humane endpoints. A humane endpoint is the earliest point at which the animal can be removed from the study that is compatible with the scientific goals of the research [3]. Humane endpoints should avoid the moribund state (i.e., a clinical irreversible state that will inevitably lead to death) and death should not be used as an endpoint. Endpoints can be developed from a generic list, but they should be tailored to the events of the specific study. Ideally, endpoints should take into account the cumulative experiences of animals on that study, for example, repeated in vivo anesthesia and imaging on a chronic tumor study as well as pain and sickness occurring with disease progression and tumor metastasis. With experience using a particular model and through the use of systematic reviews, it may be possible to refine the endpoints for certain models to surrogate endpoints or those that are less invasive [29,30,31]. Even for animals that are more challenging to monitor, for example, fish and reptiles, sensitive endpoints can be developed that indicate declining clinical condition. The humane endpoints can be discussed during the retrospective review and revised, if needed, to be more in line with the actual consequences of the project on animal well-being. Setting appropriate humane endpoints and ensuring that these are respected during the research are important means of refining research with animals and limiting cumulative suffering. As will be discussed in the following section, having appropriate tools to holistically assess animal welfare is critical for recognizing changes in animal condition and establishing needed endpoints.

## 4. Tools for Assessment of Animal Welfare

Establishing meaningful and useful limits on lifetime use and cumulative suffering for animals goes beyond the basic need for humane endpoints on any given study. This necessitates the use of periodic animal welfare assessments to evaluate potential effects on animals. Similar to an animal’s health record, welfare assessments should be conducted at regular intervals to document the potential cumulative insults to animal welfare over the duration of their life, however long or short [32]. Welfare assessment results should be reviewed by the veterinary and research team, as well as by the AEC. Moreover, as a research animal ages or declines in clinical condition, it is increasingly important to discuss their quality of life in addition to means to mitigate or end their potential suffering. This information can ultimately be used by an AEC to establish cumulative endpoints for various models or colonies of animals used in research, teaching, and production.

### 4.1. Animal Welfare Assessment Technique

Before beginning any welfare assessment, indicators to score must first be identified. Categories typically include (1) appearance, including body, coat and skin condition (e.g., unkempt coat, porphyrin staining); (2) body functions (e.g., reduced food intake, changes in body temperature); (3) environment, specifically within an enclosure where the animal is kept (e.g., nest quality, consistency of feces); (4) behavior, including social interactions, posture, gait, stereotypies; (5) procedure-specific indicators (e.g., tumor size in cancer studies, vocalization with castration); and (6) free observations in which the observers can enter their own text to describe indicators of suffering that were not pre-identified [33,34]. The CCAC specifically recommends inclusion of appropriate indicators from Table 4 below when developing welfare assessment tools for mice [35] and rats [36].

Qualitative and quantitative measures can be used independently or together to assess an animal’s welfare. The different approaches do not replace each other but rather supplement each other for a more multidimensional welfare assessment.

Quantitative assessments can refer to measures of physiologic status, such as body temperature, heart rate, and plasma or fecal levels of cortisol/corticosterone, which can all be interpreted as indirect measures of welfare. They can also refer to objective measurements of specified animal-based behavioral observations. Quantitative assessments produce data that can be used to monitor changes in welfare over time, but caution should be taken if using this data to compare welfare between animals or groups of animals. This is an excellent approach because having repeated measurements makes it possible to track changes and manage animal welfare. However, these techniques are often time consuming, costly, and dependent on the animal and environment. Unfortunately, quantitative assessments tend to be impractical for use as a routine method, such as on-farm welfare assessment. That said, this is the foundation of the widely used Welfare Quality Protocol system for pigs, cattle, and chickens [37].

The following represents a hypothetical example of a qualitative welfare assessment approach for a dog colony being maintained for drug and device discovery. At this facility, the institutional cumulative endpoint policy includes a numerical scoring system to define cumulative endpoint criteria. An overall welfare score is generated using the parameters of clinical health, procedural severity, environment quality, and behavioral assessment. The scoring system uses a value of 10 to represent the poorest welfare status and a value of 0 to represent the best possible welfare. This institution requires a welfare score to be calculated at least semi-annually (or more often, as needed) and a lifetime maximum welfare impact score of 80 is used as the criterion for retirement/adoption (Table 5).

As a secondary measure to limit severe negative welfare impact, protocols are additionally given a severity score using 10 to represent the highest negative potential welfare impact and 0 to represent no negative potential welfare impact. The potential severity score of the protocol is adjusted retrospectively, and used, along with any applicable drug washout period, to determine a rest period that is applied between successive protocols. A lifetime limit to the number of protocols based on potential protocol severity ranking is assigned and dogs are permitted to be used on one protocol ranking in the very high negative potential welfare impact category (scoring 9–10) and a maximum of one protocol ranking in the high negative potential welfare impact category (scoring 8).

Qualitative assessments evaluate the quality of animal behavior or emotions. Qualitative Behavioral Analysis is a ‘whole-animal’ approach asking human observers to summarize animals’ expressive demeanor and its context into descriptors such as relaxed, anxious, content or frustrated. These terms appear to have direct relevance to animal welfare [38]. Qualitative assessments use words, rather than scores, indices and measurements, to assess the animal’s welfare. While these assessments can be recorded, it makes it difficult to compare findings between different time points or between animals.

Incorporation of both quantitative and qualitative parameters to assess the overall procedural impacts should be used together to capture physiologic pain (e.g., injury, inflammation) and mental distress or psychological impact. It is also important to prioritize outcomes and welfare indicators at the beginning to avoid confusing interpretations later. An example of this pitfall is a recent study that relied on the use of histopathology and inflammatory and stress markers to examine the welfare impact of various blood sampling techniques in mice [39]. Because there was no prioritization of welfare indicators, no method emerged as superior even though it was clear that blood collection sites on the face or head evoked a stronger stress response whereas blood collection sites on the tail and hind limbs caused greater tissue damage.

Caution must also be taken in the interpretation of objective data as the potential effect of experimental variables must be additionally factored into any determination of the overall welfare impact. A study examining the welfare impact of 30 consecutive daily IP injections in rodents [40] involving an experienced veterinarian to administer the daily IP injection of saline concluded that multiple IP injections do not cause any ill effects in mice. Clearly the welfare impact of such procedures is influenced by concurrent variables such as the injector’s skill, the physicochemical characteristics of the test article (e.g., volume, pH, temperature, osmolality) injected and any negative effect caused by the drug (e.g., local pain, nausea, anxiety). It is important not to over-generalize the findings of this one study and assume that it is generally applicable across all situations in all rodents.

Defining specific criteria and accurately assessing animal welfare is commonly fraught with difficulty. While animals are generally accepted to be somewhere on the continuum of good to poor welfare [41], it requires an in-depth knowledge and close evaluation of behavior, physiology, psychology, and life stage of a given species to accurately assess. Furthermore, most laboratory species are in a more constrained research setting instead of a natural environment. This presents animals with opportunities to adapt and shape their natural behaviors to allow them to cope with stressors and change. As a result, special care should be taken when assessing and evaluating animal behavior in the research environment. While some generalities can be made, flexibility is necessary to account for an individual’s preferences and uniqueness. There are currently a few different welfare assessment approaches and tools that have been developed for either spot or continual assessment over time. These are described below.

#### 4.1.1. The Five Domains

The Five Domains approach has been developed and refined for assessment and management of animal welfare. While this approach has been primarily used for livestock and companion animals [42,43,44,45,46,47], it can be applied to research animals. As illustrated in Figure 4, this approach summarizes the physical/functional domains (nutrition, environment, health, and behavior), which can be positive and/or negative for an animal, which then feed into the mental domain as positive and negative affects or feelings. The overall affective experience in the mental domain equates to the welfare status of the animal.

Ideally, this Five Domains welfare assessment tool would be used on a regular interval (e.g., quarterly) to assess animals for changes in their welfare state. While this provides a snapshot to follow welfare status over time, it does not provide a measure of the cumulative stressor effects on welfare for a given animal(s). Further development of the model is needed to take into account the temporal effects and cumulative severity of animal experiences.

#### 4.1.2. Extended Animal Welfare Assessment Grid

One tool that does take time and cumulative severity into consideration is the extended animal welfare assessment grid (EWAG [8]). The EWAG system is an extension of the animal welfare assessment grid (AWAG [48,49,50]) and uses a volumetric assessment of set criteria based in spider plot analyses [51]. The welfare criteria of interest are the *x*- and *y*-axes and each criterion are then scored best (1) to worst (5). Connecting the areas over time (*z*-axis) creates a volumetric estimate (Figure 5). The smaller the resulting volume, the less the suffering and the better the welfare status is for an animal. The AWAG software is an open source collaborative project that is freely available via https://github.com/PublicHealthEngland/animal-welfare-assessment-grid/wiki (accessed on 21 April 2021). The initial idea of a volumetric welfare assessment was extended to account for time to give a more complete illustration of changes to the welfare of a given animal over time and thus cumulative impact or severity [8]. While this approach is thorough, it can be cumbersome and unwieldy to use in real time evaluation of animals [52]. Additional research and development are needed to generate additional tools that are easy to customize and practically apply.

#### 4.1.3. Biomarker Analysis

The use of biomarkers to assess animal welfare and cumulative stress has also been proposed. Telomere length is a suggested biomarker based on the premise that negative experiences (e.g., pain, injury, distress) can ultimately lead to inflammation, cortisol production, and oxidative stress at the cellular level, all of which contribute to decreased telomere length [53,54]. This effect has been demonstrated for a variety of species including fish, mammals, and birds [54,55,56,57,58,59,60,61,62,63,64]. While the utility of this approach is still widely unknown, it demonstrates promise and should be considered for additional investigation as an effective welfare assessment tool.

#### 4.1.4. Quality of Life Assessments

Quality of life (QOL) is viewed as the stability of well-being of an animal over a period of days to weeks. Periodic QOL assessments have been recommended for research primates [65], but could also be applied to other laboratory species. Simply put, a QOL assessment evaluates the balance of pleasant and unpleasant affective states during (a period of) an individual animal’s life [66]. A commonly used categorical scoring system is seen below in Table 6. The animal can move up or down in the QOL scale as the balance of positive and negative experiences changes over their lifespan. When using this approach, it is important to collaborate and seek input from all members of the research/care team, as people have familiarity with different aspects of an animal’s behavior, genetics, health, and psychological state [65]. While this approach can be holistic and easy to use, it is not very specific and can be subject to observer bias and fatigue over time. As such, this tool is best saved until an animal is approaching the end of life based on the use of other more targeted welfare assessment tools.

An example follows for how thresholds and QOL assessments can be used to define cumulative use for horses maintained in a veterinary college instructional program. In this example, the institution uses a cumulative impact policy that limits the number of interventions in a specific time period and defines a maximum total time period for use. The veterinary students learn to perform the following procedures using the teaching horses: physical examination, rectal palpation, venipuncture, venous catheter placement, and naso-gastric intubation. The institution’s cumulative endpoint policy ranks the interventions by predicted welfare impact: mild (physical examination), moderate (rectal palpation, venipuncture, and venous catheter), and marked (naso-gastric intubation). The institution may upgrade welfare impact to moderate or marked if complications are noted on retrospective evaluation (e.g., repeated number of attempts for venipuncture, blood on rectal examination). Horses are limited to a specific number of interventions in each welfare category in a week: for example, 10 mild, three moderate, one marked (evaluated on a case-by-case basis). Teaching horses are also provided periodic rest periods (e.g., one week for every four weeks of teaching use). Periodic welfare assessments are used to verify that horses can continue to be used and horses are permitted to remain in the teaching colony for a maximum of 4 years. A QOL Committee consisting of the course instructor, a husbandry staff member, AEC member, and a clinical veterinarian discuss each horse quarterly regarding their QOL and potential need for early retirement.

Hawkins and colleagues have made recommendations for welfare assessments in research animals [67]. The focus of these recommendations laid out general principles for more objective observation of animals, recognizing and assessing indicators of pain or distress, and tailoring these to individual projects. It also provided examples of systems to record welfare indicators, including score sheets and observation regimens to help detect evidence of animal suffering. Lastly, this document highlighted the importance of engaging all staff in monitoring animal welfare, from husbandry staff to veterinary staff and scientists.

#### 4.1.5. Applying Technology to Animal Welfare Assessments

Technologies for identifying infringements on animal welfare are under constant development and becoming more mainstream. Use of current and future technology will assist with increasing reliability, consistency, and efficiencies associated with identifying and monitoring health indicators. Current technologies include smart caging and digital biomarkers of welfare via radio frequency identification (RFID) into home cage activity monitoring [68,69,70,71,72,73], or home cage high-throughput imaging including thermography [74,75,76], and bioacoustics [77,78,79,80,81,82,83]. Additionally, devices such as PainTrace^®^ can be attached to an animal for monitoring of indicators of pain (e.g., nociceptive withdrawal reflex, ECG, parasympathetic tone activity) during routine procedures or surgery [84,85,86,87,88]. All these technologies allow for remote detection of changes in animals over time, removing the confounding effects of human presence or handling as well as observational bias. The data coming from these systems can all be collected and evaluated electronically, creating a trackable database of welfare indicators that can be used to evaluate cumulative severity and potentially identify cumulative endpoint criteria.

The use of smart tracking for monitoring welfare and health concerns in research animals is being used but is poorly documented in the literature. Electronic monitoring and data management have been used extensively for production animal management since the 1990′s [89,90,91,92,93] and its use has spread to both wildlife and companion animals [91,94,95]. Advances in data security continue to develop medical records [96] and research [97]. Several collaborative tracking tools with varying degrees of security are available, which are customizable, including Smartsheets (https://www.smartsheet.com/, accessed on 21 April 2021), OneDrive (https://onedrive.live.com, accessed on 21 April 2021), Google Drive (https://www.google.com/intl/en_in/drive/, accessed on 21 April 2021) and Microsoft Teams (https://www.microsoft.com, accessed on 21 April 2021). These tools can also be used across various electronic devices and platforms. Although additional research and development is necessary, these technologies could be easily applied to the research animal field to allow for easy and secure collaborative documentation of animal health and behavior for assessment of potential humane endpoints or cumulative severity experienced by a given animal [91,92]. Access to the data, raw or analyzed, may also be shared with the AEC to help with compliance-required tracking and streamline post-approval monitoring.

## 5. Recommendations for Cumulative and Lifetime Use of Animals Maintained for Teaching, Testing, and Research

### 5.1. Institutional Policy to Assess Cumulative Endpoints

Institutional policy to define cumulative welfare impacts and any resulting actions need to be established. These policies should aim to generate an ongoing summation of the welfare impacts and resultant welfare status of animals throughout their time at the research facility. Cumulative welfare impact measures should take into account the actual harms, including pain and distress, experienced by animals.

These policies should be developed by key stakeholders, such as the veterinarians, animal care and research personnel, and other members of the AEC. Administration of the cumulative welfare assessment and endpoint programs should be delegated to the AEC, which has overall authority for the institutional animal care and use program. Policies should define the protocol scoring and welfare assessment processes, as well as the resultant actions needed when animals reach a specified threshold. The interval for required assessments and the process for administration and implementation should be detailed.

When possible, institutions should incorporate refinement of procedures to reduce cumulative impact, such as routine habituation, low stress handling techniques, desensitization, and positive conditioning coupled with gold standard personnel training programs. Institutions may elect to have these positive welfare practices influence the cumulative impact score, for example by reducing the overall welfare impact score. Refinement of training programs must also be considered to reduce the cumulative impact of animals used in these programs. The use of inanimate models or cadavers for skills development in the initial phases of personnel instruction and mentor-assisted training programs will help to reduce negative welfare impacts for animals as well as overall animal use.

### 5.2. Determination of Threshold Criteria

A challenge in cumulative endpoint determination is to define the threshold level of negative welfare that is considered unacceptable. As described above in Section 4, determination of a welfare assessment and a QOL score for research animals may be used to quantify overall welfare status. Institutions will need to generate their own guidelines and incorporate details of their specific research programs as well as their institutional ethos.

## 6. Conclusions

Based on our survey results, there is widespread interest by individuals working in laboratory animal science to employ some method for determining cumulative endpoints for animals held long term or used repeatedly for teaching or research projects. Institutions have begun to address this concept although approaches tend to be informal and may not be applied consistently in all situations or across all species.

The many permutations of how and why research animals may be kept and worked with make it challenging for regulatory and compliance agencies, AECs, and institutions to develop specific policies that will cover all eventualities. While decision making should be as objective as possible, there will always be a measure of subjectiveness when scoring qualitative variables. Implementation of standardized welfare assessment tools has been used in other animal sectors to improve decision making for animals. To be widely used and implemented, tools that are developed need to be simple, inexpensive, and broadly accessible to the research community around the world. Because these assessments may generate large amounts of data over time, tools that interface with common software platforms are preferred. Further research into and development of specific and robust welfare assessment tools are needed to support institutional planning and recommendations surrounding cumulative endpoints for research animals.

## Figures and Tables

**Figure 1 animals-11-02031-f001:**
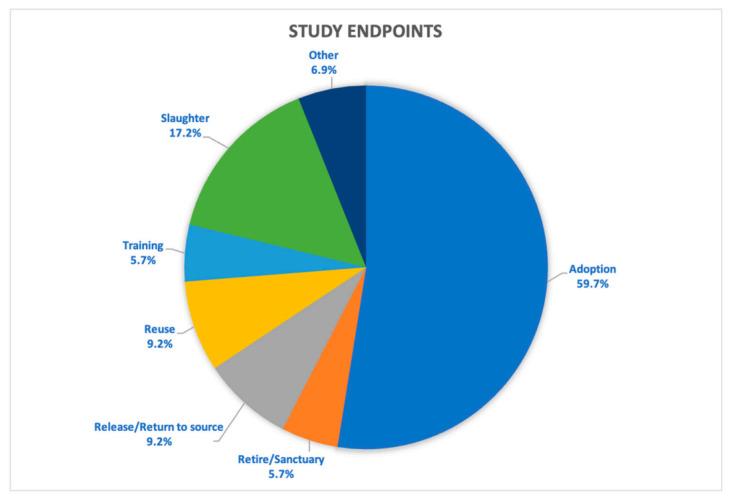
Responses to options available at study endpoint other than euthanasia.

**Figure 2 animals-11-02031-f002:**
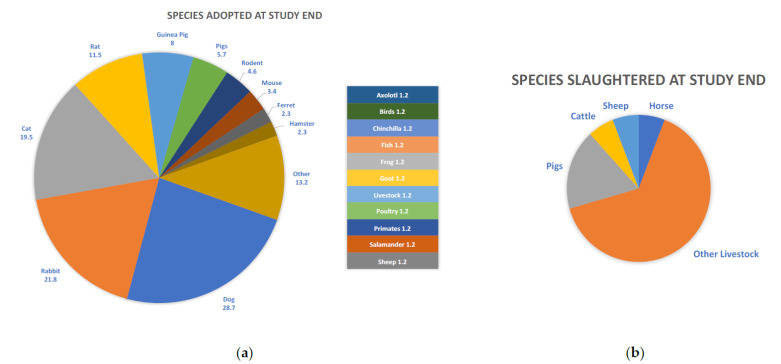
Species most likely to be considered for endpoints other than euthanasia, including (**a**) adoption and (**b**) food production.

**Figure 3 animals-11-02031-f003:**
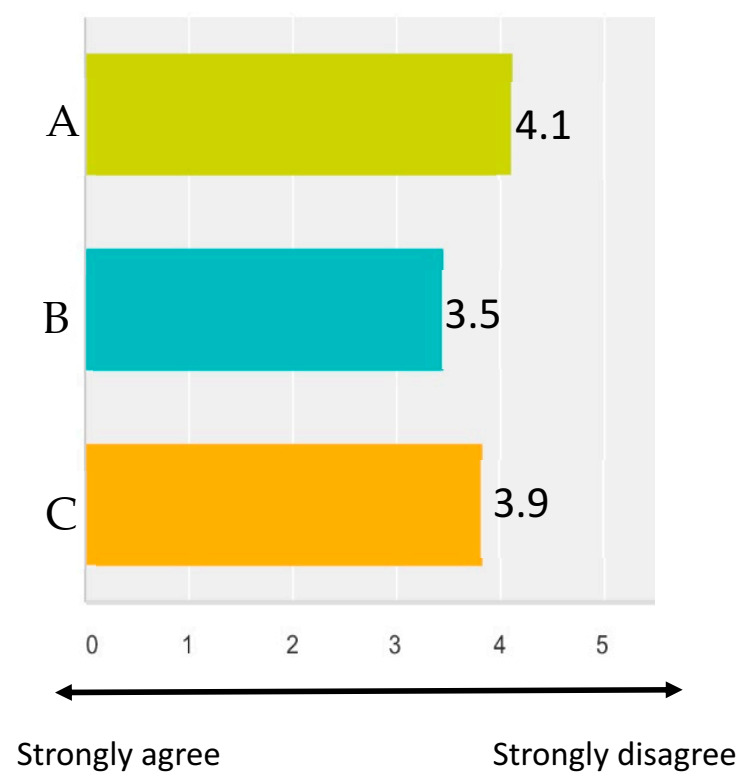
Scenarios regarding the reduction and cumulative use of research animals. (**A**) Animals of any species should not be removed from the research program, regardless of use, as this necessitates purchasing new replacement animals (i.e., contradicts reduction principle). (**B**) Provided that there is no physical trauma, such as rectal mucosal tears, it is acceptable for students to perform rectal examinations on teaching mares twice weekly for the mare’s entire institutional life (e.g., 10 + years). (**C**) As long as the maximum blood volume guidelines are adhered to, it is acceptable to perform an unrestricted number of venipunctures on a rat in a pharmacokinetic study. 0 = Strongly agree, 5 = Strongly disagree.

**Figure 4 animals-11-02031-f004:**
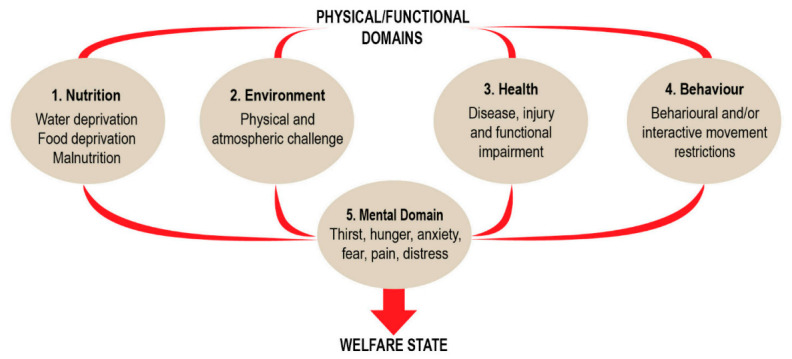
The Five Domains Model. Each of the four physical/functional domains feed into the mental domain as either positive or negative. It is the balance of the cumulative positive and negative effects on the mental domain that contributes to and determines the welfare state of the animal, from [47].

**Figure 5 animals-11-02031-f005:**
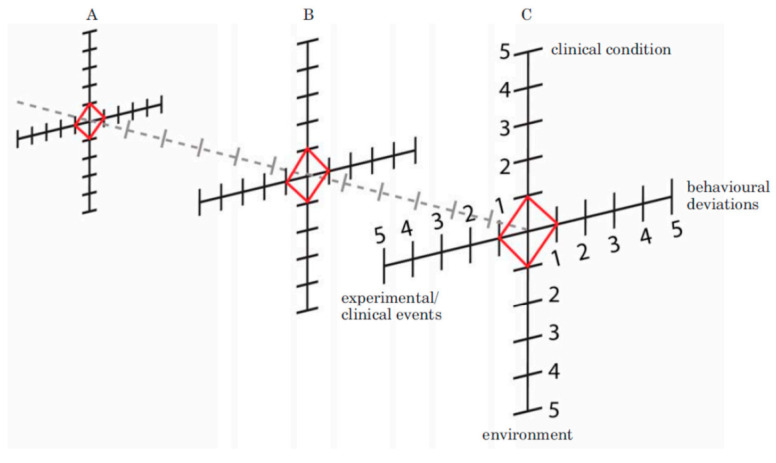
Extended Welfare Assessment Grid. Provides a volumetric assessment of animal welfare over time for a given animal. Figure represents welfare assessments performed at three different time points (**A**–**C**) spaced 6 time units apart on the time axis (dashed line). The volume resulting from connecting the three areas generated with each welfare assessment represents the cumulative severity experienced by that animal. Reprinted from [8] with permission.

**Table 1 animals-11-02031-t001:** Demographics of survey respondents.

Question	Response	% of Respondents
Geographic region	USA	50.0
	E.U.	24.0
	Canada	9.7
	Asia	3.9
	Latin America	0.7
	Other	22.7
Age range	18–25	0
	26–35	27.5
	36–45	27.5
	46–55	25.5
	56–65	16.9
	Over 65	2.6
Gender	Male	33.3
	Female	66.7
Institution type	University/College	60.5
	Industry	5.3
	Contract Research Organization	3.9
	Research Hospital	7.2
	Government	9.8
	NGO or Non-Profit	4.6
	Sanctuary	0
	Other	8.6
Primary job function	Veterinarian	58.2
	Veterinary Technician	5.9
	Animal Care	5.9
	Researcher	3.3
	Compliance	4.6
Species worked with	Mouse	77.7
	Rat	68.8
	Rabbit	58.0
	Pig	57.1
	Non-Human Primate	44.6
	Fish	41.1
	Dog	36.6
	Poultry	26.8
	Cat	25.8
	Other Rodent	40.2
	Livestock	38.4
	Teaching Animals	34.8
	Other Birds	22.3
	Other	18.8

**Table 2 animals-11-02031-t002:** Institutional AEC considerations for quality of life and lifetime use endpoints.

Considerations	Responses (%)
My institutional AEC has a guidance document for determining animal disposition at study end that includes the use of objective metrics, such as number of studies conducted with animal, number of needle sticks, number of study days, protocol invasiveness, total blood volume provided over a lifetime, etc.	36.3
My institutional AEC has established adoption and/or retirement criteria for a species, regardless of study use (e.g., based on animal age, time in facility, etc.).	25.0
My facility has an assessment protocol for scoring animals exhibiting aversive behaviors to procedures (e.g., vocalization during blood collection, excessive trembling, struggling).	31.8
My institutional AEC periodically discusses quality of life of aging and long-term housed research and teaching animals.	73.6
Animal care and veterinary personnel at my facility regularly review quality of life and endpoint decision making for aging and long-term housed research and teaching animals.	18.7

**Table 3 animals-11-02031-t003:** Examples of protocol areas impacting cumulative endpoints.

Area of Protocol Review	Example	Possible Suggestions for Protocol Authors by AEC Reviewers
Multiplicity of procedures	Teaching animals in an animal science or veterinary medicine program	Consideration for inanimate models for skills development—use to replace some or all of procedures. When live animals are a required component of training programs, consideration of maintaining higher numbers to permit using animals at lower frequency (as well as inclusion of refinements to reduce welfare impact).
	Colony or herd animals that are pooled for potential research use, e.g., ponies and horses maintained as a research herd	Define a maximum time that animals can remain in herd or colony. Retire or rehome animals once this timeline has been met.
	Scarce or special resource animals maintained as a colony and reused on studies	Clear treatment plans for maintenance of quality of life based on health conditions. Define a maximum number of studies an individual can be enrolled in. Define maximum number of minor surgeries. Define maximum number of anesthetic events. Define a lifetime maximum number of protocols in the moderate and severe welfare impact categories.
Frequency of use	MRI imaging of animal with tumors under general anesthesia 2 x/week for multiple weeks	Try to limit imaging to most critical time points, provide extra food treats and high energy foods to offset weight loss from fasting.
	Repeated studies of blood sampling for pharmacokinetic analyses	Habituate animals to bleeding, train for voluntary blood collection, when possible, counter condition animals with food treats, define a maximum period of time that animals are maintained in PK colony, rehome or adopt at study end.
Intensity of use	Pharmacokinetic study with 12 time points in 24 h	Discuss expected PK results and test article half-life to determine most critical times. Ensure maximum blood volumes are not exceeded. Encourage microsampling and use of peripheral veins and replacement of fluids. Consider catheterization (temporary or permanent) for some/all of blood collections. Discuss effects of repeated or prolonged restraint stress.
	Collecting body weights of rodent pups daily for first 2 weeks of life	Encourage surrogate forms of monitoring for pup wellness, such as presence of milk spot, and body color.
	Dosing animals with test article 2 or >times daily	Train animals to procedures and voluntary ingestion, if possible, counter condition with special resources, treats, and human interaction time, look for possible refinements, such as use of mini-pump, to avoid repeated handling/restraint stress.
Duration of use	Multi-year protocols with fistulated cows	Establish clear goals for studies and determine duration of housing for animals that is consistent with current industry practices.
	Blood donor animals kept to support research colonies or for teaching or clinical use at veterinary colleges	Attempt to establish donor animals living in local communities, determine clear duration periods if animals must live in colony/clinic setting, adopt or rehome at end of duration. Define maximum number of donations based on animal personality and response to handling and blood collections over time.
	Chronic toxicology studies	Have well defined humane endpoints, ensure personnel are trained to recognize, ensure robust behavioral management program is in place (e.g., exercise, food foraging and other resources, and positive human interaction time). Define a maximum number of studies an individual can be enrolled in.
	Aged rats used in multiple studies to assess longevity therapeutics	Establishment of a program that can evaluate ongoing welfare impacts so that these can be monitored and limited. A concurrent welfare assessment program should additionally be considered and incorporated into the intervention and endpoint determinations.

**Table 4 animals-11-02031-t004:** Indicators to be considered for inclusion in welfare assessment tool generation for mice and rats. Modified from the CCAC guidelines: mice and rats [35,36].

Consideration	Indicator
Is the environment appropriate for the individual animal: resource-based measures	Environment allows physical performance of important natural behaviors
	Provision of appropriate housing and husbandry
	Presence of negative environmental features that might impair welfare
General animal-based indicators of stress, illness, pain or discomfort	Altered food and water intake
	Weight change
	Altered posture
	Altered grooming behavior
	Coat condition
	Chromodacryorrhea
	Damage to the fur or skin
	Abnormal repetitive behaviors
	Altered social behavior
	Altered activity levels
	Partially closed, sunken, or dull eyes
	Altered interactions with humans
	Altered physiological parameters
	20 kHz vocalization
	Fecal corticosterone
Animal-based indicators of neutral or positive welfare states	Exploratory behavior
	Grooming
	Play
	50 kHz vocalizations
	Nest building and time to integrate into nest test (TINT test) and nesting consolidation test scoring
Indicators for assessing welfare in specific contexts	Facial grimace scale
	Composite pain score
	Burrowing task
	Gait score
	Cornering behavior

**Table 5 animals-11-02031-t005:** Example score sheet for an individual dog from the discovery colony. The score for any criteria ranges from 0 (best possible welfare) to 10 (poorest welfare). The scores at individual time points vary but they eventually accumulate to cross the threshold for cumulative endpoint criteria that was preset at 80.

Date	Clinical Health	Procedural Severity	Environmental Quality	Behavioral Assessments	Score	Total Score
1/1/20	0	0	4	4	8	8
4/1/20	0	6	2	4	12	20
7/1/20	4	2	8	4	18	38
10/1/20	2	8	8	6	24	62
1/1/21	2	0	2	8	12	74
4/1/21	2	0	2	4	8	82

**Table 6 animals-11-02031-t006:** A quality of life (QoL) scale where the different categories are defined in terms of the relative balance of positive and negative experiences animals may have. Modified from [43].

Category	Description
A good life	The balance of salient positive and negative experiences is strongly positive. Achieved by full compliance with best practices advice well above the minimum requirements of codes of practice or welfare.
A life worth living	The balance of salient positive and negative experiences is favorable, but less so. Achieved by full compliance with the minimum requirements of code of practice or welfare that include elements which promote some positive experiences.
Point of balance	The neutral point where salient positive and negative experiences are equally balanced.
A life worth avoiding	The balance of salient positive and negative experiences is unfavorable but can be remedied rapidly by veterinary treatment or a change in husbandry practices.
A life not worth living	The balance of salient positive and negative experiences is strongly negative and cannot be remedied rapidly so that euthanasia is the only humane alternative.

## Data Availability

Dataset available from authors upon request.

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
