# Peer review of "Developing Recommendations for Cumulative Endpoints and Lifetime Use for Research Animals"

_animals, 2021, doi:10.3390/ani11072031_

Round 1

Reviewer 1 Report

I think this is a useful and practical contribution to the Three Rs literature. I make some suggestions in the attached for a clear flow from your survey through to your recommendations.

Author Response

General comments

This paper will be an extremely useful and practical addition to thinking on the three rs and the practical application of the three rs. It is well written and a pleasure to read.

-Thank-you, we greatly appreciate your comments!

It a long read though and would benefit form clear linking throughout the parts of the paper to clearly demonstrate that your survey showed a need for the establishment of better endpoints, and then onto what those endpoints should be/how they should be assessed.

-Thank-you, we have revised the links, as suggested, and agree that this improves the flow.

It would be useful to also comment on whether the survey and your recommendations are applicable globally. You occasionally mention US and EU standards, and elsewhere comment that there are no standards, for example.

-Clarified that this is a global issue in the abstract and at the end of the introduction and conclusion.

Can you make any comment, from your survey, on why other species are not covered by formal endpoints – why is the focus on mice etc. This is a just a suggestion.

-We think this is because the majority of respondents worked with mice (78%), while other species were less common (i.e., based on institutional use of animals). A statement has been added to the results to reflect this hypothesis.

It’s good to see the focus in the welfare indicators on behavioural endpoints. I might be missing it, but I think these are not picked up in your tables eg Table 2, and it is glaringly obvious. You could perhaps use Table 2 as an example of how behavioural indicators of welfare are underused and should be developed. For example, in the mare rectal exam example, the mare’s behaviour in response to approach by handlers could be a clear indicator of her tolerance and therefore whether she needs some intervention.

-Thank-you for your comment. We strongly agree and ‘behavioral’ has been added to the list of indicators for determining experimental endpoints. It is present in the AW assessment tool section, too.

Specific comments

Title

It could be useful to reflect your focus on lifetime use of animals, as well as cumulative endpoints, for this paper, if you haven’t already

-Thank-you for your suggestion. We have incorporated ‘lifetime use’ into the title, as suggested.

Results

It would really help to have subtitles for each Part of the survey in the results. It was a little difficult to follow the transition from Part C to Part D to Part E.

-Subtitles added, as suggested.

There is no statistical analysis of the results, so that should be caveated somewhere if it’s not intended to do this.

-We have added a sentence in 2.2 Data Analyses to account for this. Aggregate responses preserve anonymity of respondents but reduce the types of statistical analyses available to interrogate the data.

Line 105 replace comma with semi-colon to read “…decision-making; Partc C: lifetime…” for consistency with other components in this list

-Replaced, as suggested.

Line 288 What does ‘informal survey’ mean? Are you referring to your won study as informal, and what is the difference here between formal and informal?

-Thank-you, this comment has been deleted.

Discussion

Section 2

Lines 288-290 The last sentence of this section could be developed to lead into your next sections. Eg “In the remainder of this paper, we outline currently available guidance, and review animal welfare indicators that can be used to determine better endpoints, and make recommendations for research organisations to establish better endpoints.”

-added, as suggested.

294 Can you set this statement in a geographical context -eg global standards, or regulatory guidance in the areas in which those surveyed work, or ‘as far we are aware’, etc: “Although there is currently no specific regulatory guidance available for determining

-added the major Western countries/regions, as suggested. While we are certain this void is global, we have not inspected all documents in native languages in all countries. The CCAC is close to issuing guidance in this area.

293 cumulative endpoints for research animals,…”

-added, as suggested

319 Can you explain where Table 3 came from – is it from the literature or how was it sourced?

-this is a hypothetical table of examples, based on the authors’ collective experiences. We have indicated ‘hypothetical examples in the text to clarify.

Table 3 Insert period/full stop after Second sentence in the third column of first row, “When live animals are a required…”

-added, as suggested

433 and elsewhere – is CCAC expanded somewhere so readers know what it refers to? Ditto USDA and other acronyms used.

-defined in text at first use, as suggested.

558-563 The reference to Hawkins et al here is not specific to QOL and could usefully be referred to earlier in this section.

-thank-you, we agree, and have used the paper in the context of summing up the welfare assessment tool discussion for the preceding section.

Section 4

This section seems a useful review of tools available for welfare assessment, but I’m not clear if it is covering the same ground as pervious reviews including those referred to in your paper. It could be worth considering whether to separate it out into a different paper if it is novel, as it lengthens this paper and takes the focus away from cumulative lifetime assessment and endpoints. Alternatively, you could shorten it and clarify its relationship to the preceding material to ensure it is clearly linked to the outcome of your survey, and your recommendations and final conclusion.

-we believe that our approach in bringing these different tools together and discussing them in the context of research animal assessments is unique, but the reviewer is correct in that there are standalone papers that discuss each tool in great detail. Because the specific literature for welfare assessment of research animals is scant, and good tools are needed for making appropriate lifetime use/endpoint decisions, we think it is important to provide an overview of common tools in this paper. A linker sentence has been added to the end of Section 3 to clarify the discussion of welfare assessment tools.

Section 5

The title of this section mixes up cumulative use, frequency of use and lifetime use. Is it worth being consistent with your terminology throughout the paper including its title? It would be useful to have an introductory paragraph here that brings together your survey results and your preceding discussion and notes that you therefore have made x recommendations

-thank-you, we have edited the overall title of the paper, so this section is more consistent now

5.3 The examples in 5.3.1 and 5.3.2 are good but are distracting in this recommendations section and should go in relevant parts the paper above this section.

-thank-you, these have been moved into relevant places in Section 4

Reviewer 2 Report

The submitted review is of high importance to the field of animal research/training and a much-needed introspective look into the consideration for managing animal welfare as a holistic approach despite the lack of regulatory mandates concerning Cumulative Endpoints. The review will be utilized by AECs and AVs to provide strong support for institutional incorporation of Cumulative Endpoints if not already implemented, while refining practices for existing programs by considering new approaches and including additional species. The information is well written, organized, and summarized appropriately. 

No major concerns were identified.

Some minor comments:

  • Figure 2a – Smaller percent species between pigs and dogs are not labeled. Would be nice if either extensions with species (similar to Fig 1) were included or a list appeared in the legend comprised of survey question #7 species not appearing in graphic.
  • Figure 2b – Would state orange section as “Other Livestock” as the listed species – pigs, sheep, cattle, horses are also considered livestock.
  • Line 235 (“The approach should be a whole animal assessment by a cross-disciplinary team.”) reads similarly to line 270 (whole animal assessments should be conducted by a cross disciplinary team). Did the authors participate in the survey results? Would suggest including an affirming statement or denial statement in Section 2.1 Survey Methods.
  • Line 304, need to complete the statement – “reviewed by _?_ [13]”
  • Need to define abbreviation upon first use: Line 344, CCAC
  • Line 570 – unsure what (Do 2020 CM) is in reference to smart caging – Could not find any Comparative Medicine articles on the subject published in 2020.
  • Line 636 – example states ‘semiannually’ which by most organizations means every 6 months. However, in table 6 the dog is reviewed 5 times from 1/1/2020 to 1/1/2021. Would change to ‘quarterly’ or adjust table so that all examples agree.

Author Response

No major concerns were identified.

Thank-you for your comments.

Some minor comments:

  • Figure 2a – Smaller percent species between pigs and dogs are not labeled. Would be nice if either extensions with species (similar to Fig 1) were included or a list appeared in the legend comprised of survey question #7 species not appearing in graphic.

-revised, as suggested

  • Figure 2b – Would state orange section as “Other Livestock” as the listed species – pigs, sheep, cattle, horses are also considered livestock.

-revised, as suggested

  • Line 235 (“The approach should be a whole animal assessment by a cross-disciplinary team.”) reads similarly to line 270 (whole animal assessments should be conducted by a cross disciplinary team).

-we thought this recommendation/quote was worth repeating and we have clarified in the discussion

  • Did the authors participate in the survey results? Would suggest including an affirming statement or denial statement in Section 2.1 Survey Methods.

-added, as suggested

  • Line 304, need to complete the statement – “reviewed by _?_ [13]”

-thank-you, this is the format used by this journal for referencing papers

  • Need to define abbreviation upon first use: Line 344, CCAC

-defined, as recommended

  • Line 570 – unsure what (Do 2020 CM) is in reference to smart caging – Could not find any Comparative Medicine articles on the subject published in 2020.

-apologies, this was a pre-submission comment that has now been removed

  • Line 636 – example states ‘semiannually’ which by most organizations means every 6 months. However, in table 6 the dog is reviewed 5 times from 1/1/2020 to 1/1/2021. Would change to ‘quarterly’ or adjust table so that all examples agree.

-clarified to indicate: at least semi-annually (or more often, as needed)

Reviewer 3 Report

Excellent quality. Minor grammar/spelling errors noticed:

Line 302: "by and AEC" should be by an AEC?

Line 597:  Confusing heading. Grammar?

Author Response

Excellent quality. Minor grammar/spelling errors noticed:

-Thank-you for your comments

Line 302: "by and AEC" should be by an AEC?

-thank-you, corrected, as noted

Line 597:  Confusing heading. Grammar?

-modified to include lifetime use (in title, as well)